# TRIM Proteins and Antiviral Microtubule Reorganization: A Novel Component in Innate Immune Responses?

**DOI:** 10.3390/v16081328

**Published:** 2024-08-20

**Authors:** Charlotte Vadon, Maria Magda Magiera, Andrea Cimarelli

**Affiliations:** 1Centre International de Recherche en Infectiologie (CIRI), Univ Lyon, Inserm, U1111, Université Claude Bernard Lyon 1, CNRS, UMR5308, ENS de Lyon, F-69364 Lyon, France; 2Institut Curie, CNRS, UMR3348, Centre Universitaire, Bat 110, F-91405 Orsay, France

**Keywords:** TRIM, cytoskeleton, microtubule, interferon, virus, infection

## Abstract

TRIM proteins are a family of innate immune factors that play diverse roles in innate immunity and protect the cell against viral and bacterial aggression. As part of this special issue on TRIM proteins, we will take advantage of our findings on TRIM69, which acts by reorganizing the microtubules (MTs) in a manner that is fundamentally antiviral, to more generally discuss how host–pathogen interactions that take place for the control of the MT network represent a crucial facet of the struggle that opposes viruses to their cell environment. In this context, we will present several other TRIM proteins that are known to interact with microtubules in situations other than viral infection, and we will discuss evidence that may suggest a possible contribution to viral control. Overall, the present review will highlight the importance that the control of the microtubule network bears in host–pathogen interactions.

## 1. A General Introduction to Microtubules

When viruses reach the cell cytoplasm, they face a crowded environment and a very intricate network of roads, the cytoskeleton. The cytoskeleton is composed of three different types of polymers: actin, intermediate filaments, and microtubules. Overall, they play an architectural role that maintains the cell shape and allows it to respond to external stimuli. For the purposes of this review, we shall concentrate on MTs.

### 1.1. Microtubules

Microtubules are present in all eucaryotic cells, where they exert numerous essential functions. They contribute to cell-shape maintenance, cell motility, and the distribution of intracellular components, they serve as rails for the intracellular transport of organelles and vesicles, form the spindles separating chromosomes in mitosis and meiosis, and they are core components of cilia and flagella. MTs are built from dimers of conserved α- and β-tubulin proteins, which assemble head-to-tail into protofilaments that then associate laterally to form a hollow tube: the microtubule. MTs are thus polar polymers, as their two ends are different: the end exposing α-tubulin is the minus (−) end, while the end exposing the β subunit is called the plus (+) end. β-tubulin contains a pocket to accommodate GTP, and only GTP-tubulin can be incorporated into MTs. Once in the MT lattice, GTP is hydrolyzed to GDP + Pi, which increases the probability of MT disassembly or depolymerization. MTs thus oscillate between phases of growth and shrinkage, a behavior known as dynamic instability [1], and most of the dynamicity happens on the MT-plus end (Figure 1). In interphase cells, MT-minus ends are clustered around the MT organizing center (MTOC), while the plus ends explore the cytoplasm and extend towards the cell periphery. The main MTOC is the centrosome, but in differentiated cells, such as neurons, hepatocytes, or muscle cells, MTs can also nucleate from the Golgi apparatus.

One of the essential roles of MTs in cells, especially important for viral infection, is to serve as rails for intracellular transport. They allow the distribution of intracellular components, organelles, and vesicles to the desired location in the cells.

### 1.2. Microtubule-Associated Proteins (MAPs): Molecular Motors

MTs in cells interact with many proteins, called MT-associated proteins (MAPs). There are several families of MAPs: molecular motors transporting cargos on MTs, MAPs binding to MT lattice and controlling MT stabilization (reviewed in [2]), MT-plus-end tracking proteins (+TIPs) regulating the plus-end dynamics (reviewed in [3]), and severing enzymes, which cut MTs (reviewed in [4]). MAP localization, binding, and activity can destine MTs for different functions, and it can also regulate the architecture of the MT cytoskeleton in cells.

The class of motile MAPs, or the molecular motors, are cellular machines that transport cellular cargoes to desired locations in cells. There are two main families of motors: dynein and kinesins. Cytoplasmic dynein moves towards MT-minus ends, whereas kinesins move towards the plus ends of MTs, and the two motor families ensure bidirectional transport using the polarity of MTs. They both use ATP hydrolysis to power their movements. While there are more than 40 different kinesin proteins, organized in 14 families [5], transporting preferentially certain cargos to specific cellular destinations, there is only one cytoplasmic dynein [6] transporting cargos towards the MT-minus ends and the cell center. Processive cytoplasmic dynein is composed of two huge multiprotein complexes, dynein-1 and dynactin, and a linker protein, which connects the motor complex to the cargo. The choice of the adapter protein defines the cargo to be transported, explaining how a single motor can ensure all the (−) end-directed traffic in cells.

### 1.3. Regulation of the MT Cytoskeleton via the Tubulin Code

But how are a variety of MAPs and motors attracted to different MTs and regulated? One mechanism that allows for addressing MAPs to specific MTs is to distinguish MTs based on the tubulin code. The tubulin code englobes the use of different tubulin isotypes to build MTs and a plethora of post-translational modifications (PTMs) that can be added to tubulin molecules incorporated into MTs (reviewed in detail in [7]).

Regulating MT properties by changing their tubulin isotype composition is a process that requires the disassembly of existing MTs and the polymerization of MTs containing this novel isotype, and it is thus dependent on the dynamics of MTs (Figure 1). On the other hand, modifying existing MTs via the enzymatic addition of PTMs can be achieved almost instantaneously. Tubulin PTMs include on/off signals such as the acetylation of lysine 40 of α-tubulin, detyrosination (the removal of the ultimate tyrosine encoded by many of the α-tubulin genes), or ∆-2-tubulin (α-tubulin lacking two C-terminal amino acids) and poly-modifications, which add amino-acid chains of different lengths to the tubulins, and which include polyglutamylation and polyglycylation.

Probably the most studied PTMs are tubulin acetylation and detyrosination. Detyrosination has been shown to control the interactions of MTs with several MAPs, which can impact MT dynamics. It inhibits, for instance, the binding of the cytoplasmic linker protein CLIP170 or dynactin subunit 1 (DCTN1) to MTs, which results in reduced MT growth and persistence [8,9,10]. Detyrosination also inhibits the active depolymerization of MTs by depolymerizing motors such as kinesin-13 that acts preferentially on tyrosinated MTs [11]. In neurons, tubulin acetylation controls the directionality of the transport events [12], but there is no evidence that it affects the motor proteins themselves. 

Acetylation and detyrosination are well-studied modifications commonly used as markers of stable MTs in cells (Figure 1). However, while these modifications are often found in long-lived MTs, there is no evidence that they influence MT dynamics. Whereas acetylation has been shown to increase the flexural rigidity of MTs, thus making acetylated MTs more resistant to mechanical stress, no effect of tubulin detyrosination on MT stability parameters has been found. The current consensus is that tubulin detyrosination and acetylation accumulate on long-lived MTs, rather than that these PTMs confer stability to modified MTs.

## 2. Use and Abuse of the MT Network due to Viruses: Three Examples

As MTs represent the major axes of transportation in the cell, it is not surprising that viruses have learned to take advantage of them, either by simply using MTs or by reorganizing them in a manner that is essentially pro-viral. Without the pretense of being exhaustive, we will focus here on three different types of interactions between viruses and the microtubule network.

### 2.1. The Human Immunodeficiency Type I Virus (HIV-1) and the Use of MTs to Reach the Nucleus during the Early Phases of the Viral Life Cycle

HIV-1 belongs to the family of *Retroviridae,* and its genome is constituted via a single-strand RNA of positive polarity flanked by two long terminal repeats (LTRs). Upon its entry into target cells, this genome is reverse-transcribed into double-strand DNA (viral DNA) via the viral reverse transcriptase, and vDNA is then integrated into a stable form into the host genome (referred to at this stage as pro-viral DNA) (reviewed in detail in [13]). This gymnastic, unique in the virus world, already begins when the virion particle leaves the producing cell, but it is in the target cell that this process speeds up and is completed [14,15,16,17,18]. Upon the fusion of the viral and cell membranes, the viral cores contained within the particle (also referred to as viral nucleoprotein complexes, VNCs, or viral capsids) are released into the cell cytoplasm. VNCs are not only the structures in which reverse transcription occurs but also the means through which the viral genome is shielded from the surrounding environment and is transported to its destination: the nucleus. Early lines of evidence demonstrated that HIV-1 VNCs track and move in a bidirectional manner along MTs, suggesting the use of dynein and kinesin molecular motors [19,20]. While several studies, including from our lab, did not reveal a major role for the cytoskeleton in some key steps of the early phases of infection (for instance, the kinetics of reverse transcription and integration), these studies relied on the use of prolonged nocodazole treatment, which is extremely disruptive to the cells and may, therefore, not have allowed a full appreciation of the role of MTs under more physiologically relevant conditions [21,22,23].

Since then, numerous laboratories have obtained data in support of the importance of MTs for the early phases of HIV-1 infection and have enriched the panorama of MAPs contributing to HIV-1 cores’ migration towards the nucleus, thus behaving as pro-viral cellular factors (Figure 2) [24,25,26,27,28,29,30]. The silencing of several components of the dynein and dynactin complexes (the dynein-heavy chain DYNC1H1, the dynactin subunits ACTAR1A and DCTN2/3, and the complex adapter BICD2) results in a decrease in HIV-1 infection, underlying the importance of these complexes in HIV-1 transport [25,26]. HIV-1 capsids can associate directly with the N-ter and C-ter domains of the BICD2 adapter [25], although a recent preprint study suggested that HIV-1 could bind directly to dynein-intermediate chain and -light chains (LC8 and TcTex1) without the need for the BICD2 adapter, at least under in vitro conditions [24]. In addition, the HIV-1 capsid has been described as possessing an end-binding protein 1 (EB1)-like homology region that enables it to bind directly to CLIP170, thus allowing a more direct loading on MTs [31].

Concerning anterograde transport, kinesin-1 (KIF5A/B) has also been shown to exert a positive effect on HIV-1 infection [28], and the kinesin complex adapter protein Fasciculation and Elongation Factor Zeta 1 (FEZ1) has been reported to interact with HIV-1 viral capsids [29]. At first glance, this may be surprising, given that kinesins mediate anterograde movements and, therefore, move away from the nucleus. However, several kinesins, including those mentioned above, are important to resolve conflicts between colliding motors that move in opposite directions on the same MT. Therefore, the positive effects of these kinesins on HIV-1 infection are likely due to the removal of obstacles along the MT highway.

### 2.2. The Human Cytomegalovirus (HCMV), or the Use of the MT to Induce Mechanical Constraints in the Nucleus

A member of the *Herpesviridae* family, HCMV is a DNA virus that possesses one of the longest DNA genomes among viruses (approximately 235 kb). Upon entry, viral nucleocapsids reach the nucleus, where the linear genome circularizes, and viral transcription begins. In the nucleus, HCMV generates viral replication compartments (RCs) that eventually coalesce to form large structures. In the cytoplasm, the generation of virion particles follows the formation of a viral production center referred to as the viral assembly compartment (vAC), which is embedded in the Golgi [32,33,34,35]. The Golgi is an important microtubule organization center (MTOC) juxtaposed with the nucleus, and recent studies have evidenced an interesting regulatory link between vAC and RC that relies on the reorganization of MTs in the cell cytoplasm to stimulate transcription changes that occur inside the cell nucleus.

During infection, vACs promote the expression and recruitment of different +TIPs to enable microtubule nucleation [36,37]. Indeed, forty-eight hours post-infection, the EB-independent +TIP-transforming acidic coiled-coil protein 3 (TACC3) mRNA expression levels are increased, and the protein is later recruited to the vAC. In turn, TACC3 recruits the colonic and hepatic tumor overexpressed gene (chTOG), a protein involved in microtubule dynamics and spindle formation, overall promoting microtubule growth [36]. This virus-induced phenomenon leads to a physical rotation of the nucleus associated with the accrued accumulation of viral DNA in RCs. This virus-induced nucleus rotation involves the Linker of nucleo-skeleton and cytoskeleton (LINC) proteins Sun1 and Nesprin through an interaction with the dynein/BICD2 complex. Inside the nucleus, these external forces create intranuclear polarity, segregating viral DNA from host DNA [38] and promoting viral transcription (Figure 3). In this case, MT reorganization governs the connection between vAC present in the cytoplasm and nuclear RC by controlling the transcriptional activity and increasing the production of viral proteins that feed the vAC.

### 2.3. Vaccina Virus (VV), Fine-Tuning MT Restructurations According to the Virus Life Cycle Stage

Vaccinia virus (VV), a member of the Poxviridae family, is a large DNA virus encoding for more than 200 proteins. As early as thirty minutes after virus–cell membrane fusion, early genes start to be transcribed in the cell cytoplasm, a phase that is followed by intermediate and late waves of transcription for up to 48 h. New particles assemble in viral factories near the MTOC, requiring the dynein–dynactin functions. Two intermediate virion particles can be distinguished: Intracellular Mature Virions (IMVs), which are the precursors to some virions migrating through the trans-Golgi, and Intracellular Enveloped Virions (IEVs), which have acquired a second double membrane. IMVs can assemble without MTs, but MTs are required for IEVs’ transport to the Golgi.

In a phenomenon that can be observed as early as five hours post-infection, the MT network undergoes a profound restructuration with MTs that emerge in the absence of a discrete MTOC in a disordered and unoriented manner (Figure 4) [35]. At this time, centrosome functions are also disrupted by the virus. A global MT retractation from the periphery to a localization near the nucleus is also observed with global MT + end shrinking and the loss of EB1 comets [39]. Such early modifications are distinct from the ones observed later during the viral cycle (at 8–10 hpi) when the virus stimulates the production of actin tails protruding from the cell that are used to propel individual virion particles at the exterior of the cell. This complex orchestration of MT dynamics to accommodate the different stages of viral infection is mediated via several MT-associated proteins coded by the virus itself, such as the viral proteins A10L and L4R that bear MAP-like properties and allow the direct binding of viral cores to MTs, at least in vitro [35]. Some report shows MTs resistant to nocodazole treatment in infected cells, suggesting that a part of the network is stabilized by VV infection [39]. In this sense, very recently, A51R has been reported to bind directly to tubulin, stabilize MTs, and promote their polymerization in vitro, and overexpressing a mutant that loses MT location significantly decreases replication in macrophages [40]. 

Additional viral proteins have been described to interact with MAPs or MTs, notably for IEV transport. For instance, A36R is a membrane-associated protein of IEVs that can bind the N-terminal Tetratricopeptide Region of the Kinesin Light Chains (KLC-TPR) [41]. F12 and E2, which are instead not anchored to membranes, have been described to interact with kinesin light chains, too, and they are necessary for the efficient transport and export from IEVs [42]. E2 is the partner mediating the interaction with KLC2 preferentially [43], and both of them enhance A36R’s association with KLC1 by removing scaffold proteins covering the TPR site [44].

## 3. Can the Cell Reorganize Its MT for Antiviral Purposes? The Example of TRIM69

The examples presented above describe three of the many different manners in which viruses can seize the MT network for infection to occur efficiently. However very little is known about whether the cell can, in turn, modify its MT to fight against viral infection. Hints that this is the case were provided in the recent description of TRIM69 as a potent and broad antiviral protein that protects the cell by driving a profound reorganization of the MT. Below is a description of how this occurs.

### TRIM69

TRIM69 was first identified as a gene expressed in the spermatids of mouse testis in 2001, and it was initially named Testis-Specific Ring Finger (or Trif, not to be confounded with the Toll/IL-1 receptor (TIR)-domain-containing adapter-inducing interferon-beta gene that is involved in TLR signaling) [45]. For several years, only a few studies dwelled on the functions of TRIM69, which was reported to play a role in apoptosis during Zebrafish brain development, or in tumor control by blocking metastasis pathways [46,47,48]. It was not until 2018, almost two decades after its discovery, that the antiviral properties of TRIM69 began to be highlighted [49,50,51]. 

TRIM69 belongs to the C-IV subgroup of the family that contains well-known antiviral factors such as TRIM5alpha, which recognizes retroviral capsids in a species-specific manner, leading to their destruction [52,53,54].

The first report of its antiviral functions indicated that TRIM69 could inhibit the replication of the Dengue virus (DENV), a positive-strand RNA flavivirus [51]. The presence of TRIM69 has been reported to lead to a decrease in viral RNA replication. Interestingly, the proposed mechanism suggests that TRIM69 could interact with and degrade NS3. This finding is of interest for several reasons. First, most TRIM family members are E3-ubiquitin ligases, and this activity is important for their functions. Second, NS3 is an RNA helicase that plays various important roles during the viral life cycle, such as assisting vRNA replication as a partner of NS5 [55], cleaving the viral polyprotein as a partner of NS2B [56,57,58], and participating in cellular immune responses inhibition [59,60,61]. Two subsequent studies identified TRIM69 following genetic screens for modulators of the replication of the Vesicular Stomatitis Virus (VSV), a negative-strand RNA virus of the *Rhabdoviridae* family [49,50]. One of the studies that focused on VSV also examined the susceptibility of DENV to TRIM69 but failed to highlight an antiviral activity. As a result, the relationship between DENV and TRIM69 remains controversial [50]. Even though TRIM69’s underlying inhibition mechanism was not investigated, both studies pointed to an interaction between TRIM69 and the viral phosphoprotein (P). A functional connection between P and TRIM69 was evidenced first by co-immunoprecipitation and second by the fact that adaptive mutations in P arise following the passage of VSV on in TRIM69-expressing cells. However, in this case, TRIM69 was not reported to drive P degradation, raising the question of the importance of the E3-ubiquitin ligase activity in the antiviral properties of TRIM69.

A further step towards the comprehension of the mechanism of action of TRIM69 came from our lab following the identification of TRIM69 as a negative modulator of HIV-1 infection in IFN-I-stimulated macrophage-like THP-PMA cells through a screen that combined both functional and evolutionary approaches [62]. In this study, we showed that TRIM69 was endowed with broad antiviral inhibition properties, as it was able to inhibit HIV-1, as well as several primate lentiviruses (SIV and HIV-2): VSV, as reported in the two above-mentioned studies, and SARS-CoV-2. Collectively, these viruses represent very distinct viral families: retroviruses and negative- and positive-strand RNA viruses. The exact step of inhibition varied according to the virus, but it exerted, in general, a minor effect (in the case of VSV) to no effect at viral entry per se. In the case of HIV-1, the defect imparted by TRIM69 mapped at reverse transcription with an effect that was evident in all viral DNA intermediates. In the case of VSV, TRIM69 strongly impaired the initial round of viral RNA transcription, called primary transcription and this defect was then amplified during overall RNA replication. Concerning SARS-CoV-2, a defect in RNA replication is observable upon 6 h post-infection. In addition, a filamentous distribution of TRIM69 had been reported in one study, but the nature of this cellular distribution was not investigated [49]. Instead, we determined that TRIM69 binds to MTs and induces a profound reorganization of the MT network with the accumulation of stable microtubules (Figure 5). In support of this finding, TRIM69-decorated MTs resist nocodazole treatment and accumulate increased levels of two post-translational modifications: the acetylation on K40 and detyrosination both of α-Tubulin. Using in vitro purified components, we were able to determine that TRIM69 binds directly to MTs. Although the exact domains involved in this interaction have not yet been identified, by examining the behavior of the five described isoforms of TRIM69, which lack varying and consistent portions of the protein, we could highlight the strict relationship that existed between the ability of TRIM69 to induce MT reorganization and its ability to inhibit viral infection. Thus, our data point to the fact that TRIM69 may drive an antiviral response that makes use of MTs as a weapon against viral infection. In the broader context of IFN responses, we have demonstrated that MT reorganization can also be observed in primary human myeloid cells in response to type I IFN. This finding is particularly interesting, given that TRIM69 is a prototypical ISG but is also expressed at high levels in basal conditions in this cell lineage. A recent study supported the notion that, effectively, TRIM69 binds to MTs, and the study revealed a role for TRIM69 in centrosome dynamics [63]. TRIM69 overexpression, often found in certain tumors, has been shown to drive resistance to centrosome poisoning, which is a safety mechanism that leads to cell death in normal cells undergoing an abnormal centrosome number.

A consequence of TRIM69’s expression is the accumulation of what can be defined as stable MTs. It is important to note that the definition of stable MTs is broad and that stable MTs could likely be subdivided into heterogeneous entities. In this respect, while Taxol-mediated stabilization has been shown to exert positive effects on certain viruses [62,64], TRIM69 exerts negative effects, indicating that the mention of stable MTs may oversimplify the complex heterogeneity and functional diversification of MTs.

According to the results presented above, and since it targets a broad spectrum of very diverse viruses, the model we support suggests that TRIM69 modifies the MT, and because of this modification, viral replication is inhibited. This model would account for a single mechanism with different viral inhibition outcomes and accommodate the data in our possession for the moment. But the question we would like to address here is, given that one TRIM member has been described as able to rewire MTs for antiviral purposes, are there other TRIM members that can carry out similar antiviral functions? Several TRIM members have been reported to act on the MT in contexts other than viral infection. In the next section, we shall present all of these TRIMs in the context in which they were identified and are studied, while in the final section, we will review the data that suggest that these TRIMs may also contribute to antiviral functions in manners that have yet to be completely described.

## 4. Microtubule-Associated TRIM Proteins (TRIM-MAPs)

Contrary to TRIM69, which belongs to the C-IV TRIM subfamily, all the TRIM members known to interact with MTs belong to the C-I subfamily. The first description of a member of the TRIM family as being associated with microtubules dates back from the late 1990s with the identification of TRIM18 (Midline 1, MID1) and TRIM1 (Midline 2, MID2) as MT-binding proteins [65,66,67,68,69]. Later, following bioinformatics approaches, Short and Cox identified a novel domain through which several additional TRIM members could associate with MTs [70]. This motif was referred to as the C-terminal-subgroup-one signature (COS), as it was identified in several members of the C-I TRIM subfamily (out of IX subfamilies overall) (Figure 6). The COS domain is a loosely conserved motif spanning 60 amino acids that is posited after the coiled-coil domain present on all TRIM proteins, and it can be functionally transferred to heterologous TRIM proteins, conferring MT-binding activities. As mentioned above, our laboratory has recently identified TRIM69 as an MT-binding protein, and interestingly, TRIM69 does not possess a detectable COS domain [62]. Therefore, this finding strongly argues for additional domains through which TRIM members can associate with MTs.

### 4.1. TRIM18/MID1

TRIM18/MID1 has been genetically linked to the Opitz BBB/G syndrome (OS), a monogenic disorder characterized by developmental defects that appear in the embryo’s midline and that include lip-palate–laryngotracheal clefts and an imperforate anus. In line with its important role in this pathology, MID1 is highly expressed during embryonic development; it is conserved across vertebrates [71,72], and in humans, it is present on the distal part of the short arm of the human X chromosome (Xp22. 3), where it undergoes X-inactivation. Mutations associated with OS are often, albeit not always, concentrated in the C-terminal part of the protein. They result in frameshifts that functionally delete this portion of the protein and impair its ability to bind and bundle MTs [73,74,75,76,77,78,79]. The nature of the cellular partners identified so far for TRIM18/MID1 highlights the likely possibility that this protein may be involved in numerous physiological processes that may contribute to OS, as well as to other cellular dysfunctions. Indeed, TRIM18/MID1 has been shown to associate with and dock the regulatory subunit alpha4 (α4) of PP2-type phosphatases onto MTs [80,81,82,83]. By leading to the degradation of α4, this interaction normally controls the activity of PP2 phosphatases, which are implicated in a wide variety of physiological processes. Failure to degrade α4 has been linked to decreased mTORC1 formation, S6K1 phosphorylation, and cap-dependent translation, features that can be observed in cells obtained from patients carrying TRIM18/MID1 mutations [84,85,86]. Given that all the pathogenic TRIM18/MID1 variants described are unable to associate with MTs and bundle them [66,69,87], we can speculate that the loss of the spatial regulation of α4 on MTs is an important function mediated via TRIM18/MID1.

However, the functions of TRIM18/MID1 may extend beyond the regulation of PP2 activities. There is substantial evidence supporting the potential involvement of MID1/TRIM18 in translation. Indeed, TRIM18/MID1 is associated with a translation-competent complex that includes RACK1, Annexin A2, and Nucleophosmin, as well as proteins of the small ribosomal subunits [88]. Furthermore, TRIM18/MID1 is found to be associated with a regulatory complex involved in the translation of expanded CAG repeats, which are a shared feature of different neurodegenerative disorders, such as Huntington’s disease [89]. It is thus possible that TRIM18/MID1 could play a role in the compartmentalization of the process of translation of certain mRNAs, a feature that can also contribute to its physiological functions during the embryo’s development and to the pathology related to loss of MT binding.

Finally, although these features have not been studied extensively, TRIM18/MID1 has been shown to alter T-cell motility in an mTOR-dependent manner [90,91], leading to accrued inflammation following experimental infections [89,92], and to influence the exocytosis of lytic granules from cytotoxic T cells along with TRIM1/MID2 [93,94].

### 4.2. TRIM1/MID2

TRIM1/MID2 is closely related to TRIM18/MID1, with which it shares an identical protein-domain organization. TRIM1/MID2 exhibits a somewhat broader pattern of expression when compared to the latter, characterized, for example, by a higher expression in the heart [67]. TRIM1/MID2 binds MT and can also associate with Astrin, a microtubule-organizing protein, and thus it bears the possibility to alter cytokinesis and cell division [95]. However, mutations in TRIM1/MID2 have not so far been described in OS patients. Instead, a recent study highlighted the importance of TRIM1/MID2 for the recruitment of the leucine-rich repeat kinase 2 (LRRK2) protein on MTs [96], an important finding, given that mutations in LRRK2 are a common cause of familial Parkinson’s disease (PD) [97]. The weight of this finding in PD must be confirmed, but this observation is of potential interest, as TRIM1/MID2 could act as a docking site on MTs for cellular proteins that, however, have not been identified so far.

### 4.3. TRIM9/SPRING (SNAP-25-Interacting RING Finger Protein)

TRIM9/SPRING is expressed in the brain and concentrated at cellular synapses, where it is involved in neuronal axon branching in response to Netrin-1 [98,99,100,101]. TRIM9/SPRING interacts with the Netrin-1 receptor, deleted in colorectal cancer (DCC), and normally degrades it through its E3-ubiquitin ligase activity. However, following netrin-1 stimulation, TRIM9/SPRING promotes DCC multimerization, which in turn decreases the receptor susceptibility to ubiquitination, stabilizing it and leading to the recruitment of the focal adhesion kinase (FAK). In this manner, TRIM9/SPRING promotes soluble N-ethylmaleimide-attachment protein-receptor (SNARE)-mediated exocytosis [102]. TRIM9/SPRING also interacts directly with one SNARE component, SNAP25, as well as with the actin barbed-end polymerase vasodilator-stimulated phosphoprotein (VASP), and therefore, it links SNARE-mediated exocytosis and filopodial stability to netrin-dependent axon guidance [98,99,100,101]. In support of its important role in axon guidance, neurons lacking TRIM9/SPRING exhibit several morphological defects, including excessive dendritic arborization [103]. Interestingly, TRIM67/TNL, which is closely related to TRIM9/SPRING, has been shown to compete with TRIM9/SPRING for interaction with VASP, a competition that is functionally important for netrin-1-dependent filopodial responses, axon turning, and branching [104,105].

Finally, TRIM9/SPRING has been described as repressed in the brains of individuals suffering from Parkinson’s disease and dementia with Lewy bodies [106]. While these findings remain observational, it is of potential interest that two TRIM members with MT-binding capacity (TRIM1/MID2 and TRIM9/SPRING) have been linked to Parkinson’s disease.

### 4.4. TRIM67/TNL (TRIM Nine-like)

TRIM67/TNL is highly expressed in the cerebellum, and as mentioned above, it binds its paralog, TRIM9/SPRING, with which it competes to bind to the Netrin-1 receptor DCC [105], as well as for axon guidance [104]. TRIM67/TNL has also been described to promote SNARE exocytosis by limiting the incorporation of SNAP47 into SNARE complexes [99,107]. The importance of TRIM67/TNL in brain development is underlined in the findings showing that *TRIM67^−/−^* mice present hypotrophy of several regions of the brain (the hippocampus, striatum, amygdala, and thalamus), in addition to a thinning of forebrain commissures and impairments in spatial memory, cognitive abilities, and muscle functions [105].

Several reports suggest a link, not yet fully clear, between this TRIM member and cancer. TRIM67/TNL is involved in signaling from the rat sarcoma protein (Ras) through an interaction with the p53-Responsive Gene 1 (PRG-1) and the protein kinase C substrate 80K-H (80K-H), proteins that are degraded upon ectopic expression of TRIM67/TNL, leading to an overall attenuation of cell proliferation in cell lines [108]. Not only can TRIM67/TNL interact directly with p53, which prevents its MDM2-mediated degradation, but it is also a transcriptional target of p53. Thus, TRIM67/TNL expression is likely involved in a loop of reciprocal amplification with p53 that ultimately drives the accumulation of p53 and leads to cell growth inhibition and apoptosis. In this respect, the finding that TRIM67/TNL downregulation is associated with poor survival rates in colorectal cancer is not surprising [109,110].

On the other hand, TRIM67 has been described to promote cell proliferation, migration, and cell invasion in non-small cell lung cancer (NSCLC) by positively regulating the neurogenic locus-notch homolog protein (Notch) pathway [111,112]. In addition, auto-antibodies against TRIM9/SPRING and TRIM67/TNL have been described as a potential high-risk paraneoplastic biomarker [113], and TRIM67/TNL is associated with higher cell motility and reduced cell adherence in glioma cells. These results may, overall, indicate that this TRIM member, perhaps through its role in MT regulation, can play a role in the metastatic process of several cancer types [114].

Overall, the link between TRIM67/TNL and cancer development appears to be dependent on the cellular context of expression, and perhaps on the cell environment. In this respect, it is important to note that TRIM67/TNL, undetectable in the liver under normal conditions, can be induced in this organ in conditions of obesity, where it activates hepatic inflammation, exacerbating the progress of non-alcoholic fatty liver disease (NAFLD) found in this pathology [115,116].

### 4.5. TRIM36

TRIM36 was initially cloned from the tumor suppressor gene region of chromosome 5q22.3, frequently altered in different types of tumors [117]. However, our understanding of the contribution of TRIM36 to the process of tumorigenesis remains incomplete. TRIM36 interacts with the centromere protein H (CENP-H) and delays cell-cycle progression; therefore, it occupies a prime position from which to influence the process of tumorigenesis [118]. In addition, TRIM36 interacts and suppresses via ubiquitin-mediated degradation the forkhead box factor A2 (FOXA2, [119]), a transcription factor that plays an important role in metabolism and tumor-cell proliferation. It is thus not a surprise that TRIM36 expression is downregulated in certain cancers, such as liver cancer [120], lung adenocarcinoma (LUAD [121]), esophageal squamous-cell carcinoma cells [122,123], and prostate cancer [124].

For the moment, no firm evidence links mutations in TRIM36 to a given pathology, although a genome-wide association study in a Chinese cohort indicated it as a susceptibility gene for pre-eclampsia, a condition of arterial hypertension that occurs in pregnant women and that can be hereditary [125]. Furthermore, a homozygous mutation in TRIM36 has been associated with autosomal recessive anencephaly in an Indian family [126]. In this latter case, the pathogenic mutation in TRIM36 abolishes its ability to bind MTs and alters neuronal cell proliferation.

### 4.6. TRIM46/TRIFIC (Tripartite, Fibronectin Type III, and C-Terminal B30.2-like Motif)

TRIM46/TRIFIC is highly expressed in the brain, where it is subjected to alternative splicing and where it organizes parallel microtubule arrays on the axons, a function important for neuronal polarity and axonal function [127]. Studies carried out on knockout rats underlined that TRIM46/TRIFIC plays an important role in hippocampus development and axon development by regulating the size and distance between axon initial segments (AIS). The AIS separates the axon from the soma of neuronal cells and controls, in an MT-dependent manner, the selective transport of cargo into the axon-dendritic compartment. Hence, this compartment constitutes a molecular separator, a sort of molecular filter, that is important in axon-dendrite differentiation. Mechanistically, TRIM46/TRIFIC drives the formation of closely spaced parallel microtubule bundles oriented with their plus-end outward, driving AIS structuration [128,129,130].

As many of the TRIM members mentioned above, TRIM46/TRIFIC has been linked to cancer, and it is amplified in lung adenocarcinoma (LUAD) tissues and has often been associated with a poor survival rate [131]. The underlying molecular mechanism is also, in this case, likely multifactorial. TRIM46/TRIFIC interacts with and negatively regulates the FK506-binding protein 5 (FKBP5), which is itself linked to cell activation [132,133]; it interacts with and ubiquitinates the peroxisome proliferator-activated receptor (PPAR, [134]), and it is involved in the regulation of β-catenin [135]. Lastly, auto-antibodies against TRIM46/TRIFIC have been proposed as a biomarker of paraneoplastic CNS disorders and clear-cell renal-cell carcinoma [136,137].

## 5. Reviewing Potential Links between TRIM Members Endowed with MT-Binding Abilities and Virus Infection Modulation

The TRIM family is composed of approximately 80 members in humans. Most of them can heterodimerize or homodimerize and thus can influence each other in numerous ways that can span stability, localization, and the capacity to organize in higher-order structures. One level of regulation of this capacity occurs at the level of expression with some TRIM proteins that are expressed constitutively in certain tissues, or only after specific stimulation of which interferon is the most studied, but likely not the only one, as is the case for TRIM67/TNL. This complexification is a caveat of most, if not all, studies on TRIM proteins that make the abstraction of the possibility that several TRIM members may modulate the activity of the TRIM protein studied. However, this simplification is needed not only because of the extremely high number of combinations that should be tested but also because several TRIM members are expressed in multiple isoforms. Currently, aside from TRIM69, the antiviral potential of the TRIM members endowed with MT-binding properties has not been explored to its full extent. Certain studies have highlighted the possibility that this may be the case (Figure 7), although the true weight that MT reorganization may play remains unclear. With this section, it is our hope that this evidence will raise new interest in the connections between TRIM-MAPs and antiviral responses and inspire much-needed studies in this area.

A survey of TRIM family-member expression in primary blood cells stimulated with classical type-I IFN suggests that TRIM69 may be the only TRIM-MAP expressed in these cells [138]. However, this observation should not be taken as final proof of a lack of expression because it does not exclude the possibility that certain members may be expressed only in specific cell subsets, or only after specific stimulations. On the contrary, all the TRIM proteins mentioned above are expressed in tissues of the central nervous system (CNS), where they may undergo specific regulation and where they will have the ability to influence the replication of several neuro-tropic viruses.

Several TRIM-MAPs intersect directly with the interferon-signaling pathway. For instance, TRIM18/MID1 has been shown to degrade the type-I interferon receptor 2 [139], as well as the interferon regulatory factor 3 (IRF3, [140]), while TRIM46/TRIFIC leads to decreased IFN responses by interacting with the TANK-Binding Kinase 1 (TBK1, [141]). To strengthen a potential link with inflammatory responses, several TRIM-MAPs exert regulatory and often opposite effects over the nuclear factor kappa-light-chain-enhancer of activated B cells’ (NF-κB) signaling pathway, which, of course, occupies key roles in inflammatory responses’ regulation. For example, TRIM46/TRIFIC promotes the ubiquitination and degradation of the NF-κB inhibitor IκBα, leading to the activation of the NF-κB signaling pathway [142]. In contrast, TRIM9/SPRING and TRIM67/TNL exert an opposite effect on this pathway by displacing the β-transducin repeat-containing protein (β-TrCP) from the Skp-Cullin-F-box ubiquitin–ligase complex. This results in an inhibition of IκBα degradation, driving dampened proinflammatory responses [143,144,145,146,147]. To complexify the matter, alternative isoforms that are often observed in the case of TRIM genes may exert distinct regulatory functions. This has been proven in the case of TRIM9/SPRING, in which a short isoform (TRIM9s) positively regulates type I IFN signaling by activating IRF3 signaling [148,149], thus exerting global effects opposite to the ones of the full-length protein.

As such, TRIM-MAPs can participate in IFN antiviral responses, although whether they exert a direct antiviral effector function remains to be determined, apart from TRIM69. Evolutionary analyses indicated that TRIM69 evolves under positive selection in primates and other mammals. This feature indicates that TRIM69 undergoes mutations faster than other genes, a characteristic often associated with the engagement of a cellular antiviral protein with a viral antagonist. It is the interaction between these two opposite elements that drives the need to mutate in each of them in the interfaces through which they interact: the element targeted to escape the interaction and the element targeted to regain it. While these marks highlight a possible host–pathogen interaction that occurred through evolutionary times, it does not reveal which viral lineage may have exerted evolutionary pressure on TRIM69. However, two viral proteins have been reported to physically interact with TRIM69: the phosphoprotein P of VSV and NS3 of DENV. Even though the connections between NS3 and TRIM69 remain controversial, both proteins have been described as either binding directly to MAPs or being involved in the formation of viral factories in the cytoplasm of infected cells, often juxtaposed to MTs. In addition, the P protein of VSV is part of the viral nucleoprotein complex that enters target cells upon infection, together with the viral proteins N, the polymerase L, and the viral RNA. This complex uses MTs to migrate in the early stages of infection, and dynein components are often purified along with them. The VSV nucleocapsid complex also forms intracytoplasmic bodies’ replication factories (IBs). Thus, the possibility that Flavivirus NS3 or Rhabdoviridae P proteins may interfere with or be subjected to TRIM69 regulation in a manner that intersects with the viral need for MTs is of potential interest, and it requires further investigation.

At present, a single report suggests that viral infection may change the expression levels of the mentioned TRIM members. Indeed, infection with the neurotropic rabies virus induces the downregulation of TRIM9/SPRING. While this may underline an indirect effect due to global changes in the neuron proteome, it may, on the other hand, reflect a specific interaction between TRIM9/SPRING and viral proteins that could, if this were the case, further support the argument of the existence of a specific virus–host antagonisms [150].

Finally, the expression of TRIM18/MID1 in humans is driven by an endogenous retroviral promoter. This is interesting because this feature is common to several ISGs, and it is suggested to have shaped interferon-sensitive genes at the genomic scale [151,152]. Whether this represents a fortuitous evolutionary event or whether it participates in a wider scheme that places TRIM-MAPs within the broader context of antiviral responses remains to be determined.

## 6. Conclusions

Overall, the MT network is exploited in numerous different manners by viruses to accommodate essentially pro-viral purposes, ranging from the mere usage of existing MTs to the profound restructuration of the entire network. Until recently, it remained unclear whether the cell could, in turn, reorganize its MT and use this as a novel weapon against viral infection. The identification of TRIM69 as the first member of the TRIM family that reorganizes MTs, inducing broad viral inhibition, highlights the fact that the cell bears the ability to exploit its MT to fight off viral infections. In this respect, it is of interest that several other members of the TRIM family are indeed TRIM-MAPs and are, therefore, endowed with an intrinsic ability to bind with MT and alter MT dynamics, but for the moment, these TRIMs have been essentially studied in conditions other than viral infection (genetic diseases, cancer, and nervous system development). Yet, evidence gathered through the literature indicates that TRIM-MAPs bear a high potential to alter viral infection, either indirectly by modulating IFN responses or more directly by modulating MT dynamics, as highlighted in our results with TRIM69. Lastly, the fact that TRIM69 evolves under strong positive selection in primates is highly evocative of its engagement in an evolutionarily important host–pathogen interaction, and although the virus or viruses driving it have not been identified yet, this finding strengthens the possibility that the struggle for the control of the MT is a previously unrecognized facet of host–pathogen interactions.

## Figures and Tables

**Figure 1 viruses-16-01328-f001:**
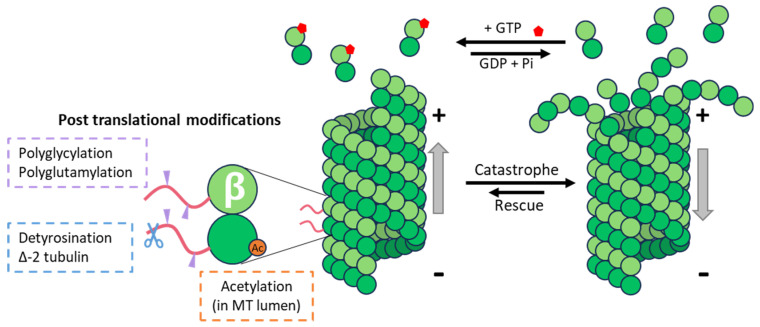
Microtubule dynamics and tubulin post-translational modifications. MTs’ (+) end undergoes dynamic polymerization, with the addition of new αβ-GTP (in red) tubulin dimers, and depolymerization. The presence of certain post-translational modifications of α- and or β-tubulin is associated with stable microtubules that are more resistant to mechanical stress and catastrophe events.

**Figure 2 viruses-16-01328-f002:**
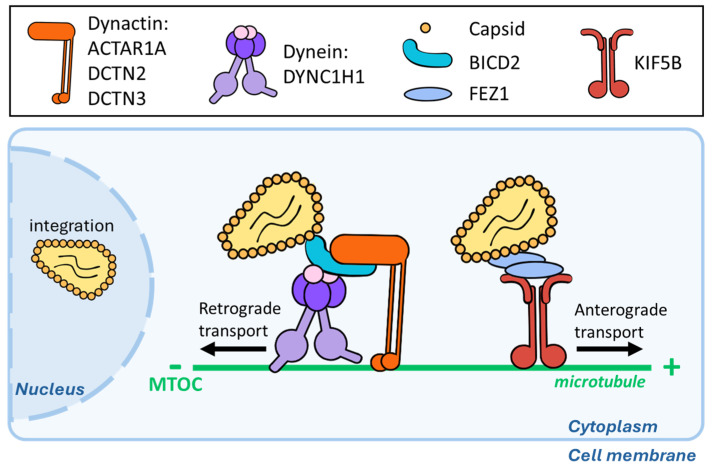
HIV-1 viral cores use MTs to reach the nucleus. HIV-1 cores have been reported to load both on dynein and kinesin complexes and to move in a bidirectional manner along MTs. The dynein adapter BICD2 binds directly to the Capsid protein, and the other different members of the complex involved in this retrograde transport are highlighted at the top of the graphic. For the anterograde counterpart, FEZ1 seems to mediate the direct interaction with the viral core.

**Figure 3 viruses-16-01328-f003:**
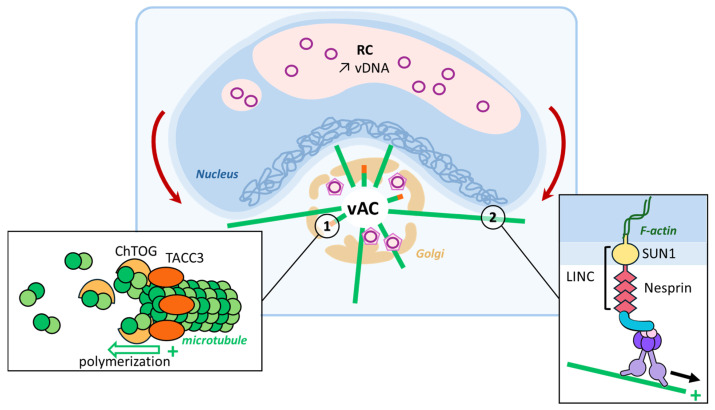
MT reorganization induced via HCMV. HCMV hijacks the Golgi apparatus and turns it into a viral assembly compartment (vAC) where virions form and acquire a second envelope (1). TACC3, the expression of which is increased upon infection, recognizes the + end of MTs and recruits ChTOG involved in microtubule elongation, thanks to its ability to bind tubulin dimers and the MTs. (2) Dynein motors present on these MTs can bind to the LINC complex anchored in the nuclear membrane. This tension, mediated via the LINC proteins on the nucleoskeleton, segregates the host genome toward the nuclear membrane from vDNA to increase viral transcription levels, and it results in the characteristic kidney-shaped nucleus observed upon infection.

**Figure 4 viruses-16-01328-f004:**
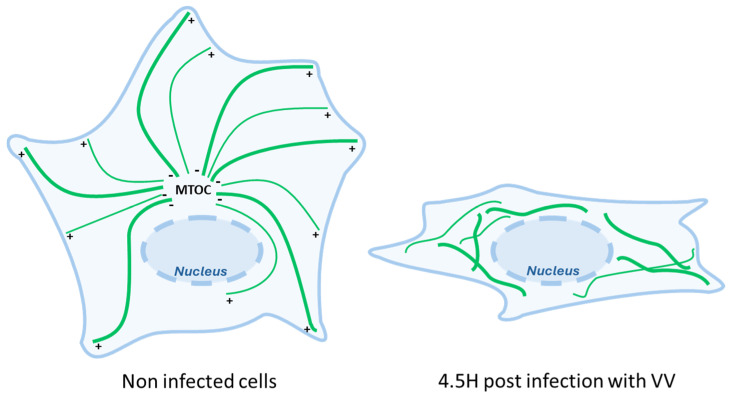
The drastic changes in the MT network induced via Vaccina virus infection. Cells infected by the Vaccinia virus undergo profound microtubule network reorganization characterized by the loss of order in MT organization and by an overall drastic retraction of MTs from the periphery toward the nucleus.

**Figure 5 viruses-16-01328-f005:**
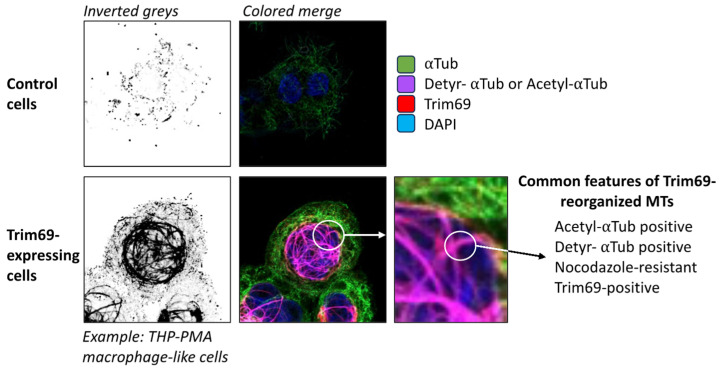
Phenotypical changes in the MT induced via TRIM69. This confocal microscopy reproduces and summarizes the effects of the expression of TRIM69 on the MT in THP-1 cells differentiated into a macrophage-like status upon incubation with PMA (THP-PMA). The results presented here present a summary of the results published in reference [63].

**Figure 6 viruses-16-01328-f006:**
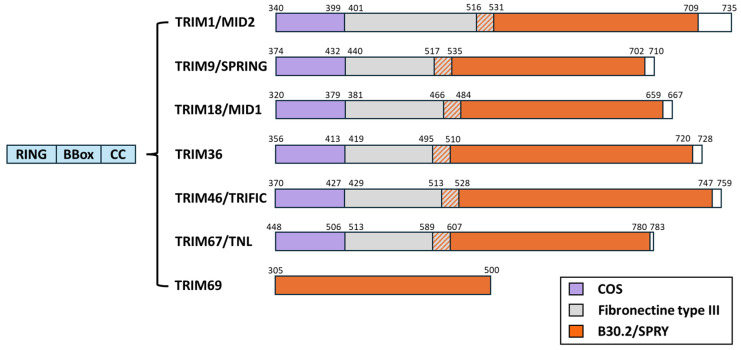
Schematic representation of TRIM proteins known, to date, to interact with the MT. As their name implies, TRIM proteins share common N-ter domains and present a RING, one or two BBox, and a coiled-coil domain (blue). The most variable portion of the proteins lies in the C-terminal. Members of the TRIM family C-I subgroup harbor in their C-terminal a COS domain (purple) that has been reported to mediate their association with MTs. These TRIM proteins also bear a Fibronectin type III domain (gray) and a B30.2/SPRY domain (orange). TRIM69 belongs to the C-IV subgroup; it does not possess a detectable COS domain, it lacks the FNIII one, and it harbors a B30.2/SPRY domain.

**Figure 7 viruses-16-01328-f007:**
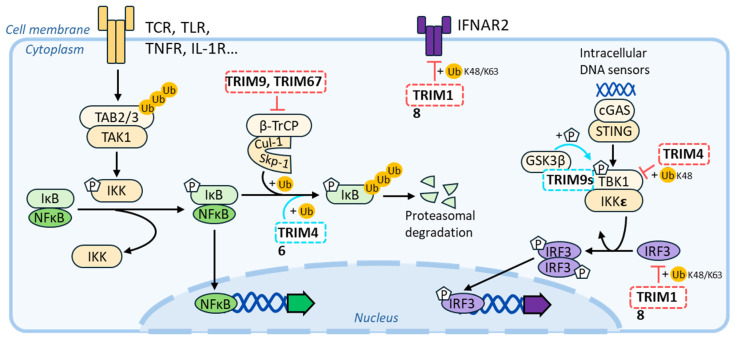
Schematic summary of the connections between MAP-TRIMs and antiviral signaling pathways. Positive and repressive effects of MAP-TRIMs on a given pathway are indicated with a blue and red color, respectively.

## Data Availability

Not applicable.

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
