# Peer review of "TRIM Proteins and Antiviral Microtubule Reorganization: A Novel Component in Innate Immune Responses?"

_viruses, 2024, doi:10.3390/v16081328_

Round 1

Reviewer 1 Report

Comments and Suggestions for Authors

Vadon and colleagues wrote a comprehensive review on microtubule (MT) binding TRIM proteins regarding their potential roles as antiviral factors. The first half of the review is very interesting which will help us understand the role of MT-binding TRIM69 in innate responses against viral infections. The latter half of the review, however, has a room for improvement. Here are several points to be addressed.

1. Although TRIM69 has been shown to have antiviral roles against various viruses, other TRIM proteins described in Section 4 have not been studied in the context of viral infection. Therefore, whether these MT-binding TRIM proteins have antiviral roles is currently unknown. Although the authors have tried to discuss a possible contribution of these proteins to viral control in Section 5, the proposed mechanisms are indirect and antiviral roles of MT-binding TRIMs are still purely speculative. Is it possible to provide convicting evidence or better implications that these TRIM proteins have antiviral roles? It would be also helpful to discuss more in detail how likely (or unlikely) these TRIM proteins contribute to innate responses to viral infection.

2. The antiviral role of TRIM69 protein can be described more in detail. For example, which replication cycle is affected by the TRIM protein in HIV, VSV, or SARS-CoV-2 infection.

3. Lines 267-269. It would be nice to clarify that although two studies (ref #49, 50) investigated the role of TRIM69 in VSV replication, one of them ALSO tested DENV2 and found no effect of TRIM69 on DENV2 replication unlike the previous paper (ref #51). The reviewer was confused by the sentence at first glance.

4. Line 278, “following co-immunoprecipitation” should be “evidenced by co- immunoprecipitation” or something like that.

Comments on the Quality of English Language

There seem some typos and unclear expressions.

Author Response

Dear Editor,

Please find the answers to the referee’s comments, which were overall highly positive. The major changes are highlighted in yellow in the revised version of the review. Typos and minor changes are instead not highlighted.

Reviewer 1

Reviewer 1 defined our review as « an outstanding treatise on the state of our understanding of the interplay between the cell and viruses in control of the MT network » and we sincerely thank them.

Reviewer 2 Report

Comments and Suggestions for Authors

Microtubules (MTs) are critical to many aspects of cellular structure and function, including maintenance of the integrity of cell shape, cell motility, intracellular transport of cellular components such as chromosomes, organelles and vesicles not to mention being primary structural components of cilia and flagella.  As such vital cellular components, it is not at all surprising that many viral pathogens have developed and perfected strategies designed to usurp aspects of the MT network for their own purposes.  In turn, of course, cells have adapted to counteract these viral strategies to overtake the MT network.  In this review, the authors initially present a general summary of the structure, function and regulation of the MT cytoskeleton, followed by a detailed review of the strategies used by viruses to use the MT network to take over the cell and the efforts of the cell to counteract these strategies and maintain and protect the cell’s structural and functional integrity.  Finally, the review delves in detail into the role of a subset of innate immune factors, Trim proteins, in the innate immune response.

This is considered an outstanding treatise on the state of our understanding of the interplay between the cell and viruses in control of the MT network.  There are a number of strengths evident in the review that raise it to an elite level.  From the outset, the introductory summary of MT structure and regulation lays a strong foundation for that which follows, presenting as it does the intricacies of MT structure, function and regulation in a clear and succinct fashion.  Due to the diversity of strategies developed by RNA and DNA viruses and retroviruses to take advantage of the MT network by converting them to essentially pro-viral entities, it was considered a particularly astute decision by the authors to use three very different examples of how viruses “use and abuse” the MT network for their own purposes.  This serves the purpose of demonstrating the diversity of viral strategies without bombarding the reader with a more all-inclusive survey across the spectrum of viruses.  To their credit, the authors admit this summary is not exhaustive.  This section is a major strength of the review.

Contrary to the overall discussion of viral strategies to usurp the MT network, the authors present highly detailed discussion of the Trim proteins, as a relatively large family of innate immune factors that use an array of different strategies to protect the cell against viral (and bacterial) attacks.  Here, that approach is appropriate, as the authors make it clear that there is no apparent limit to how the cell can adapt to counteract the viral strategies through Trim protein mediated efforts, including the ability of Trim69 to drive a significant reorganization of the MT network.  This represents a strategy where the cell actually uses the MT network as an antiviral tool.  Several additional examples are discussed that demonstrate the power of the Trim family proteins ,of which there are approximately 80.  This leads to a discussion of Trim protein structure and, in particular, the role of the COS domain in their activities.  Finally, the authors make it clear that Trim protein-related defects can be linked to a wide array of diseases, including Parkinson’s disease, dementia, Opitz, BBB/G Syndrome, pre-eclampsia, colorectal cancer and tumor metastases.  This includes the caveat that the Trim67/cancer development link can be influenced by the environment.  Also illuminating is the recognition that Trim69 can evolve under positive selection and mutate to better respond to the virus.

In summary, this review makes a strong case for the importance of the MT network as a viral target, as well as the cell’s development of strategies, especially the family of Trim proteins, to counteract the viral attacks.  It is considered a timely contribution to the field that significantly advances our understanding of the topic.

Comments on the Quality of English Language

Minor check for grammar and style needed.

Author Response

Dear Editor,

Please find the answers to the referee’s comments, which were overall highly positive. The major changes are highlighted in yellow in the revised version of the review. Typos and minor changes are instead not highlighted.

Reviewer 2

Reviewer 2 appeared also extremely positive and put forward certain suggestions, which we have taken into account. Specifically we have,

1)    deleted the basic information on the virology on DENV

  • specified that TRIM69 belongs to the subclass C-IV as other well known antiviral factors (for ex TRIM5alpha)
  • Specified in Figure 7 that the role of the MT in the described antiviral functions of the TRIM-MAPs remains to be elucidated. Overall, we hope this review will spur new interest in the subject and hopefully inspire new studies.

We have not performed positive selection analyses on other Trims, but we agree with the reviewer that this is a very interesting topic.

We have substituted TRIM in capital letters and corrected the writing throughout the manuscript.

Reviewer 3 Report

Comments and Suggestions for Authors

The review by Vadon et al works to establish the as-yet untested hypothesis that microtubule-binding TRIMs may regulate microtubule dynamics as part of their innate antiviral functions. The hypothesis is based on the well-described observation that many TRIMs have roles in antiviral defense and that a handful of TRIMs appear primarily localize to microtubules. The authors highlight their own study focused on TRIM69 that supports their hypothesis, and posit that other microtubule-localized TRIMs may also exert antiviral activities through alterations of cytoskeleton function.

As a review that highlights the actions of a subset of microtubule-localized TRIMs, this manuscript is unique in the literature. However, given the lack of publications reporting that microtubule-localized TRIMs (aside from TRIM69) have antiviral effects, it seemed to me that the review could have focused more on what is known about microtubule-localized TRIMs in general rather than focusing so much effort to discussing virology. Overall, I think the manuscript could be much shorter while still being of equal (or even higher) value.

Main points:

I feel that much of the information provided in the manuscript was extraneous to understanding the functions of microtubule-binding TRIMs. For example, in lines 255-259, the authors discuss the virology of DENV. This was unnecessary. I recommend trying to eliminate excessive text to allow the authors space to focus more on the actions of microtubule-localized TRIMs or even expand their discussion of TRIM69.

Perhaps the authors could specify that TRIM69 belongs to a sub-class of TRIMs (IV) that also includes many famous restriction factors. On the other hand, none of the COS domain-containing TRIMs (which are best known for microtubule binding) are known antiviral factors.

The authors point out that there is evidence for positive selection of TRIM69. To further strengthen their hypothesis that other microtubule-associated TRIMs may have antiviral functions, they could provide an analysis of positive selection for those genes.

Figure 7: This is perhaps the most interesting and useful figure in the manuscript. However, it is very interesting to point out that none of the TRIMs highlighted in this figure are obviously engaging microtubules in their functions (in fact, microtubules are not even in the figure).

While the writing was generally clear, there were many instances in which non-standard English was used.

In the literature, when human TRIMs are discussed “TRIM” is capitalized (not written Trim).

Comments on the Quality of English Language

There a multiple instances in which non-standard English was used. All of these are very minor, and could likely be resolved using free editing software. 

Author Response

Dear Editor,

Please find the answers to the referee’s comments, which were overall highly positive. The major changes are highlighted in yellow in the revised version of the review. Typos and minor changes are instead not highlighted.

Reviewer 3

Reviewer 3 is very positive on the review and proposes some improvements, which we have taken into account. Specifically:

  • we agree with the reviewer that at present the published evidence in favor of a role of TRIM-MAPs in antiviral modulation remains to be firmly established, we hope that by raising and discussing these possibilities, our review will inspire new studies that will address exactly this topic.
  • According to their comments, we have better specified the steps inhibited by TRIM69 in the different viruses tested
  • We have clarified the sentence on DENV